# Semi-supervisedly Co-embedding Attributed Networks

**Zaiqiao Meng**[*]
Department of Computing Science
Sun Yat-sen University, and University of Glasgow
zaiqiao.meng@gmail.com

**Shangsong Liang**[†]
School of Data and Computer Science
Sun Yat-sen University
liangshangsong@gmail.com

**Jinyuan Fang**
School of Data and Computer Science
Sun Yat-sen University
fangjy6@gmail.com

**Teng Xiao**
School of Data and Computer Science
Sun Yat-sen University
tengxiao01@gmail.com

## Abstract

Deep generative models (DGMs) have achieved remarkable advances. Semi-supervised variational auto-encoders (SVAE) as a classical DGM offer a principled framework to effectively generalize from small labelled data to large unlabelled ones, but it is difficult to incorporate rich unstructured relationships within the multiple heterogeneous entities. In this paper, to deal with the problem, we present a semi-supervised co-embedding model for attributed networks (SCAN) based on the generalized SVAE for heterogeneous data, which collaboratively learns low-dimensional vector representations of both nodes and attributes for partially labelled attributed networks semi-supervisedly. The node and attribute embeddings obtained in a unified manner by our SCAN can benefit for capturing not only the proximities between nodes but also the affinities between nodes and attributes. Moreover, our model also trains a discriminative network to learn the label predictive distribution of nodes. Experimental results on real-world networks demonstrate that our model yields excellent performance in a number of applications such as attribute inference, user profiling and node classification compared to the state-of-the-art baselines.

## 1 Introduction

Network embedding has been receiving significant attention in recent years due to the ubiquity of networks in our daily lives. The goal of network embedding is to encode entities, e.g. nodes and attributes, of a specific network into low-dimensional representation vectors, while features of the network, e.g. nodes' topological structure [1] and their communities [2], can be decoded from the inferred embedding vectors. Various graph-based applications, e.g. node classification and clustering [3, 4], community detection [2, 5], link prediction [6] and expert cognition [7], have been shown to be able to benefit from network embedding techniques.

Attributed networks as one category of the most important networks are ubiquitous in a myriad of critical domains, ranging from online social networks to academic collaborative networks, where rich attributes, e.g. nationalities of the users and journals/conferences the authors published at, describing the properties of the nodes, are available. Many embedding methods for attributed networks [8, 9, 10] have been proposed to learn the low-dimensional vector representations of nodes via leveraging their

---

[*]This work was done when the first author was with the Sun Yat-sen University.
[†]Shangsong Liang is the corresponding author.

topological structure and the associated attributes. In this paper, to further enhance the effectiveness of the learned representations, we jointly consider whatever information of the network we have in most real-world scenarios: topological structure and attributes of all nodes, and a few labels associated with some of the nodes. We refer to such kind of networks owning these information as *partially labelled attributed networks*. Embeddings trained with partially label information can be used to boost the performance of related tasks, where not all the label data are available. For instance, in tag recommendation for social network users, tags manually labelled for the expertise of some users can be utilized to train an embedding model to predict tags of those users missing their tags.

A number of works has been proposed to learn low-dimensional representations for networks either in unsupervised [8, 9, 10] or semi-supervised [11, 12, 13, 14] ways. However, both of the existing unsupervised and semi-supervised attributed network embedding methods learn representations for nodes only, and thus are not able to capture the affinities/similarities between nodes and attributes, which are the key to the success of many attributed network applications, such as attribute inference [15, 16] and user profiling [17]. Moreover, a vast majority of existing works predominately represent node embeddings by a single point in a low-dimensional continuous space, resulting in the fact that the uncertainty of nodes' representations can not be captured.

Deep generative models (DGM) have achieved remarkable advances due to their profound basis in theoretic probability and flexible and scalable optimizing ability of deep neural networks. Semi-supervised variational auto-encoders [18, 19] as a classical DGM offer a principled framework to effective generalize from small labelled data to large unlabelled ones, but have limited applicability to incorporate rich unstructured relationships within the multiple heterogeneous entities. To alleviate the aforementioned problems, we introduce a **S**emi-supervised **C**o-embedding **A**ttributed **N**etwork algorithm, SCAN, built based on the generalized SVAE for the heterogeneous data, that is able to **co-embed both attributes and nodes** of partial labelled networks **in the same semantic space**. SCAN collaboratively learns low-dimensional vector representations of both attributes and nodes in the same semantic space in a semi-supervised way, such that the affinities between them can be effectively measured and the partial labels of nodes can be fully utilized. The learned nodes' and attributes' embeddings in the same semantic space are, in turn, utilized to boost the performance of many down-stream applications, e.g., user profiling [17], where the relevance of users (nodes) and keywords (attributes) can be directly measured by, e.g., cosine similarity. In our SCAN, we infer the embeddings of both nodes and attributes and represent them by means of Gaussian distributions. With the natural property of latent presentations (i.e., Gaussian embeddings in our case), SCAN innately represents their uncertainty with the corresponding variances of the inferred embeddings.

Our contributions can be summarized as follows: [1] (1) We generalize SVAE to the heterogeneous data and propose a novel semi-supervised co-embedding model for attributed networks, SCAN, to collaboratively learn representations of nodes and attributes in the same space such that proximities between nodes as well as affinities between nodes and attributes of networks can be effectively measured. (2) Our SVAE model jointly optimizes a variational evidence lower bound consisting of five atomic observations based on two entities and two relationships, to obtain the Gaussian embeddings of nodes and attributes and a discriminator for node classification, where the mean vectors denote the position of nodes and attributes and the variances capture uncertainty of their representations. (3) We perform extensive experiments on real-world attributed networks to verify the effectiveness of our embedding model in terms of three network mining tasks, and the results demonstrate that our model is able to significantly outperforms state-of-the-art methods.

## 2    Related Work

Many unsupervised representation learning methods have been proposed to embed various networks into low-dimensional vectors of nodes. Approaches such as DeepWalk [1], node2vec [2] and LINE [3] learn embeddings for plain networks, where only topological structure information is utilized based on random walks or edge sampling. The Structural Deep Network Embedding [20] model embeds a network by capturing the highly non-linear network structure so as to preserve the global and local structure of the network. Wang et al. [5] incorporate the community structure of network into result embeddings to preserve both of the microscopic and community structures. Some other work obtains embeddings for non-plain networks with rich auxiliary information, such as labels, node attributes

and text contents, in addition to the topological structure networks [21, 22]. Hamilton et al. [10] propose the GraphSAGE model that learns node representations by sampling and aggregating features from nodes' local neighbourhoods. Zhang et al. [9] and Gao et al. [23] propose their customized deep neural network architectures to learn node embeddings, while capturing the underlying high non-linearity in both topological structure and attributes. CAN [24] is a model that unsupervisedly learns embedding for both nodes and attributes by their customized DGM. Their results show that combining different types of auxiliary information, rather than using only the topological features, can provide different insights of embedding of nodes. Recently, a few approaches embed nodes by distributions and capture the uncertainty of the embeddings [25, 26, 6]. For example, the KG2E model [25] represents each entity/relation of knowledge graphs as a Gaussian distribution. In terms of embedding attributed networks, Aleksandar et al. [6] embed each node as Gaussian distribution according to the energy-based loss of personalized ranking formulation.

Recently, learning representations of entities for networks by semi-supervised ways has also been widely studied. Planetoid [11] is a network representation learning model for semi-supervised node classification but not for capturing affinities between embeddings of nodes and attributes. Kipf et.al [12] propose a graph convolutional neural network model for attributed networks for semi-supervised classification task, which also outputs embedding for nodes only. Liang et.al [13] propose a semi-supervised learning model, called SEANO, which utilizes dual-input and dual-output deep neural networks to learn node embedding encompassing information related to structure, attributes, and labels explicitly alleviating noise effects from outliers. More recently, a semi-supervised deep generative model for attributed network embedding has been proposed [14], which applies the generative adversarial nets (GANs) to generates fake samples in low-density areas in networks and leverages clustering property to help classification. However, it still learns embeddings for nodes only. Therefore, the embeddings obtained by these models might not be directly utilized in other applications such as attribute inference and user profiling.

## 3 Semi-supervised Variational Auto-encoder

The Semi-supervised Variational Auto-encoders (SVAEs) are a kind of generative semi-supervised models for partially labelled data that learn representations of data by jointly training a probabilistic encoder, probabilistic decoder as well as a label predictive neural networks [18, 19], while the small labelled data sets can be generalized to large unlabelled ones. SVAEs have received a lot of attention due to their wide applicability in domains such as text classification [27], machine translation [28] and syntactic annotation [29]. In this section, we fist review the construction of an SVAE for homogeneous entities, and then extend it to the setting of heterogeneous data.

### 3.1 Semi-supervised Learning for Homogeneous Data

We first consider the homogeneous data that appear as pairs $\mathcal{O} = \{(\mathbf{x}_1, \mathbf{y}_1), \ldots, (\mathbf{x}_N, \mathbf{y}_N)\}$, with $\mathbf{y}_i \in \{0, 1\}^{1 \times K}$ being the one-hot vector representing the label of $i$-th observation $\mathbf{x}_i \in \mathbb{R}^F$ where $K$ is the number of classes and $F$ is the feature dimension. In order to incorporate unlabeled data in the learning process, previous work on deep generative semi-supervised models optimize a variational bound $\mathcal{J}$ on the marginal likelihood for $N_1$ labelled data points and $N_2$ unlabelled data points:

$$\mathcal{J} = \sum_{i=1}^{N_1} \mathcal{L}(\mathbf{x}_i, \mathbf{y}_i) + \sum_{j=1}^{N_2} \mathcal{U}(\mathbf{x}_j), \tag{1}$$

where $\mathcal{L}(\mathbf{x}_i, \mathbf{y}_i)$ is the evidence lower bound (ELBO) for a labelled data point and $\mathcal{U}(\mathbf{x}_j)$ is the ELBO for an unlabelled one. Normally, a SVAE model also incorporates a classification loss into Eq. 1 to impose the label posterior inference model $q_\phi(\mathbf{y}|\mathbf{x})$ to be able to act as a classifier:

$$\mathcal{J}^\alpha = \mathcal{J} + \alpha \cdot \mathbb{E}_{\widetilde{p}_l(\mathbf{x}, \mathbf{y})} \left[ -\log q_\phi(\mathbf{y}|\mathbf{x}) \right], \tag{2}$$

where hyper-parameter $\alpha$ controls the weight between generative and purely discriminative learning.

While these models are conceptually simple and easy to train, one potential limitation of this approach is that marginalization over all $K$ classes becomes prohibitively expensive for a large number of classes. We leave the discussion of the solution for this limitation to subsection 4.2.

Another limitation is that they only learn representations for the homogeneous observations, which are generated independently according to a homogeneous prior, ignoring that heterogeneous data

observations are ubiquitous in the real world. For example, in recommender system items and users are two independent observations but they are associated with purchase or interaction behaviours. Many efforts have been paid to co-embedding multiple entities for heterogeneous systems in a fully unsupervised learning procedure [24, 30]. However, to our knowledge, no model in the literature can learn embeddings for heterogeneous data with multiple entities in a semi-supervised way. Hence, in the next subsection we will present a model that allows to learn multiple entity observations having arbitrary conditional dependency structures and arbitrary label for each type of entity.

## 3.2 Semi-supervised Learning for Heterogeneous Data

We consider a generalized form of semi-supervised model for heterogeneous data that appear as triples instead of pairs. Let $\mathcal{O} = (\mathcal{X}, \mathcal{Y}, \mathcal{R})$ be a triple set of the heterogeneous observation data, where $\mathcal{X} = (\mathbf{X}^1, \cdots, \mathbf{X}^T)$ is the entity set with multiple types (e.g. $T$ types), $\mathcal{R} = \{r_{ij} \mid \mathbf{x}_i^g \in \mathbf{X}^g, \mathbf{x}_j^h \in \mathbf{X}^h\}$ is a set containing relationships with $r_{ij}$ being the relationship strength of two entities in same/different types, and $\mathcal{Y} = (\mathbf{Y}^1, \cdots, \mathbf{Y}^T)$ is the partially labelled set for all types of entities.

Without loss of generality, we can formulate the semi-supervised learning model by first considering only two entity types, e.g. $\mathbf{X}^g$ and $\mathbf{X}^h$. We let $\mathcal{O}_{ij} = (\mathbf{x}_i^g, \mathbf{x}_j^h, r_{ij}, \mathbf{Y}^l)$ be an atomic data point of the observation where $\mathbf{Y}^l$ are the labels of entities if there are, $\mathcal{Z}_{ij} = (\mathbf{z}_i^g, \mathbf{z}_j^h, \mathbf{Y}^u)$ be the collection of latent variables of the two entities where $\mathbf{Y}^u$ are the predicted labels for entities without a label, and $\mathbf{Y} = (\mathbf{Y}^u, \mathbf{Y}^l)$. $\mathcal{F}_{ij} = (\phi(\mathbf{x}_i^g), \phi(\mathbf{x}_j^h))$ is the conditional variables of variational posterior where $\phi$ is a function taking entities' feature as input for filtering posteriors. Then, the logarithm marginal likelihood of $\mathcal{O}_{ij}$ can be written as:

$$\log p(\mathcal{O}_{ij}) \geq \mathbb{E}_{q_{\phi}(\mathcal{Z}_{ij}|\mathcal{F}_{ij})}[\log p_{\boldsymbol{\theta}}(\mathcal{O}_{ij}, \mathcal{Z}_{ij}) - \log q_{\phi}(\mathcal{Z}_{ij} \mid \mathcal{F}_{ij})], \quad (3)$$

where the joint distribution, i.e. $p_{\boldsymbol{\theta}}(\mathcal{O}_{ij}, \mathcal{Z}_{ij})$, can be represented as:

$$p_{\theta}(\mathcal{O}_{ij}, \mathcal{Z}_{ij}) = p_{\theta}(\mathbf{x}_i^g, \mathbf{x}_j^h, r_{ij} \mid \mathcal{Z}_{ij}, \mathbf{Y}) p(\mathbf{Y}) p(\mathbf{z}_i^g) p(\mathbf{z}_j^h). \quad (4)$$

In most real-world scenarios, such as recommender systems [31] and network embeddings [24], people only concern about the reconstruction of relations between entities rather than the feature of the entities, thus $p_{\theta}(\mathbf{x}_i^g, \mathbf{x}_j^h, r_{ij} \mid \mathcal{Z}_{ij}, \mathbf{Y})$ can be simplified to be $p_{\theta}(r_{ij} \mid \mathcal{Z}_{ij}, \mathbf{Y})$. With the mean-field approximate inference of the variational posterior $q_{\phi}(\mathcal{Z}_{ij} \mid \mathcal{F}_{ij})$, Eq. 3 can be written as:

$$\log p(\mathcal{O}_{ij}) \geq \mathbb{E}_{q_{\phi}(\mathcal{Z}_{ij}|\mathcal{F}_{ij})}[\log p_{\boldsymbol{\theta}}(r_{ij} \mid \mathcal{Z}_{ij}, \mathbf{Y})] - KL(q_{\phi}(\mathcal{Z}_{ij} \mid \mathcal{F}_{ij}) \mid p(\mathbf{z}_i^g) p(\mathbf{z}_j^h))$$
$$\triangleq -\mathcal{L}(\mathcal{O}_{ij}), \quad (5)$$

where $\mathcal{L}(\mathcal{O}_{ij})$ is denoted as the ELBO of $\mathcal{O}_{ij}$. Note that $\phi$ can be different types depending on the heterogeneity of the observation entities. Incorporating an additional weighted discriminative component as in Eq. 2 leads to the following lower bound for overall observations:

$$\mathcal{J}^{\alpha}(\mathcal{O}) = \sum_{r_{ij} \in \mathcal{R}} \mathcal{L}(\mathcal{O}_{ij}) + \alpha \cdot \mathbb{E}_{\widetilde{p}_l(\mathbf{x}, \mathbf{y})} \left[ -\log q_{\phi}(\mathbf{y}|\mathbf{x}) \right]. \quad (6)$$

# 4 Deep Semi-supervised Attributed Co-embedding

We now turn to the semi-supervised learning problem for the partially labelled attributed networks.

## 4.1 Problem Definition

Let $\mathcal{G} = (\mathcal{V}, \mathcal{A}, \mathbf{A}, \mathbf{X}, \mathbf{Y}^l)$ be a **Partially Labelled Attributed Network**, with $\mathcal{V}$ and $\mathcal{A}$ being the sets of nodes and attributes, respectively, $\mathbf{A} \in \mathbb{R}^{N \times N}$ and $\mathbf{X} \in \mathbb{R}^{N \times M}$ being the weighted adjacency matrix and node attribute matrix, respectively, where $N = |\mathcal{V}|$ is the number of nodes and $M = |\mathcal{A}|$ is the number of attributes. $\mathbf{Y}^l$ is the label matrix representing the node labels. Since most nodes' labels are unknown, $\mathcal{V}$ can be divided into two subsets: labelled nodes $\mathcal{V}^l$ and unlabelled nodes $\mathcal{V}^u$ with their label matrix being $\mathbf{Y}^l$ and $\mathbf{Y}^u$ respectively.

The problem of **Semi-supervised Attributed Network Co-embedding** is defined as:

**Problem.** *Given a partially labelled attributed network $\mathcal{G}$, learn a mapping function $\Xi$ that satisfies the following in a semi-supervised way:*

$$\mathcal{G} = (\mathcal{V}, \mathcal{A}, \mathbf{A}, \mathbf{X}, \mathbf{Y}^l) \xrightarrow{\Xi} \mathbf{Z}^n, \mathbf{Z}^a, \mathbf{Y}^u, \tag{7}$$

*such that both network structure, node attributes and the partial labels can be preserved as much as possible by $\mathbf{Z}^n$, $\mathbf{Z}^a$ and $\mathbf{Y}^u$, where $\mathbf{Z}^n \in \mathbb{R}^{N \times D}$ and $\mathbf{Z}^a \in \mathbb{R}^{M \times D}$ represent the latent representation matrices for all the nodes and attributes, respectively, and $\mathbf{Y}^u$ represents the learned labels for all the unlabelled nodes. Here $D$ is the size of the embeddings.*

## 4.2 The Semi-supervised Co-embedding Model

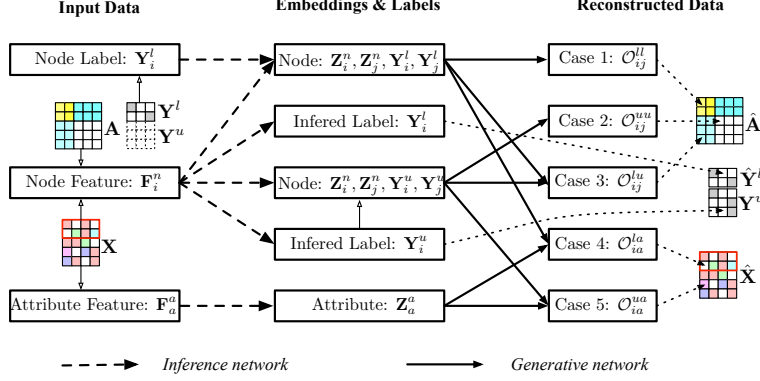

Figure 1: Our model takes the adjacency matrix ($\mathbf{A}$), the attribute matrix ($\mathbf{X}$) and the partial node labels ($\mathbf{Y}^l$) as input, and outputs Gaussian distributions as latent embeddings for all nodes and attributes ($\mathbf{Z}^n$ and $\mathbf{Z}^a$), as well as the latent labels of the unlabelled nodes ($\mathbf{Y}^u$). The two neural network models, i.e. the inference network and the generative network, are trained by optimizing the ELBO on the log marginal likelihood of the five cases of observations.

The partially labelled attributed networks are obviously heterogeneous data, as the nodes and attributes can be considered as two types of heterogeneous entities, and the adjacency matrix and the attribute matrix can be regarded as the relationships between these two entities. To address the problem, we propose the **SCAN**, a **S**emi-supervised **C**o-embedding model for **A**ttributed **N**etwork that semi-supervisedly co-embeds both attributes and nodes in the same semantic space, i.e., learns latent variables/embeddings for both nodes and attributes, allowing for effective generalization of classification from a small number of labelled data sets to a large number of unlabelled ones. In what follows, we first follow the principle of modelling semi-supervised learning for heterogeneous data (Subsection 3.2) to derive an overall lower bound for this problem by splitting the given attributed network observations into five types of atomic data points, then provide a solution for addressing the bottleneck of the marginalization over all $K$ classes. Fig. 1 provides an overview of the framework of our SCAN, and Fig. 2 provides probabilistic graphical perspective of our model.

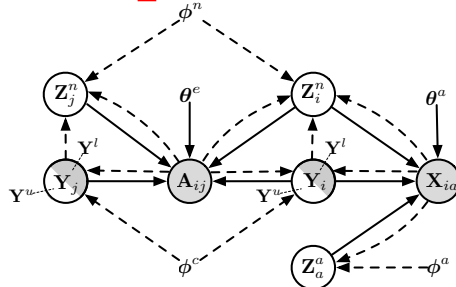

Figure 2: Probabilistic graphical model of our SCAN. Generative model dependencies are shown in solid arrows, while inference model dependencies are shown in dashed arrows.

**The Variational Evidence Lower Bound.** As shown in Fig. 1, we have two relationship observations: the adjacency matrix $\mathbf{A}$ and the node attribute matrix $\mathbf{X}$. The elements in both of matrices (i.e. the edges between nodes and the attribute values of nodes) can be categorized into five cases: (case 1) an edge connecting two labelled nodes; (case 2) an edge connecting two unlabelled nodes; (case 3) an edge connecting a labelled node and an unlabelled node; (case 4) an attribute value associating with a labelled node and an attribute; (case 5) an attribute value associating with an unlabelled node and

an attribute. We can easily obtain the ELBOs for these five types of atomic observations according to Eq. 5, namely, $\mathcal{L}(\mathcal{O}_{ij}^{ll})$ for case 1, $\mathcal{U}(\mathcal{O}_{ij}^{uu})$ for case 2, $\mathcal{M}(\mathcal{O}_{ij}^{lu})$ for case 3, $\mathcal{B}(\mathcal{O}_{ia}^{la})$ for case 4 and $\mathcal{C}(\mathcal{O}_{ia}^{ua})$ for case 5, respectively [1].

Once all the ELBOs for the above five cases are obtained, we can obtain the variational bound on the marginal likelihood for the entire adjacency matrix and node attribute matrix as follows:

$$\mathcal{J}(\mathbf{A}, \mathbf{X}, \mathbf{Y}^l) = \sum_{i,j \in \mathcal{V}^l} \mathcal{L}(\mathcal{O}_{ij}^{ll}) + \sum_{i \in \mathcal{V}^u, j \in \mathcal{V}^u} \mathcal{U}(\mathcal{O}_{ij}^{uu}) + \sum_{i \in \mathcal{V}^l, j \in \mathcal{V}^u} \mathcal{M}(\mathcal{O}_{ij}^{lu})$$
$$+ \sum_{i \in \mathcal{V}^l, a \in \mathcal{A}} \mathcal{B}(\mathcal{O}_{ia}^{la}) + \sum_{i \in \mathcal{V}^u, a \in \mathcal{A}} \mathcal{C}(\mathcal{O}_{ia}^{ua}). \tag{8}$$

In the objective function of Eq. 8, the first three terms on right-hand side are the ELBOs on the marginal likelihood of edges, while the other two terms are the bound loss for attributes. To make our model more flexible to govern the loss between the edge data points and the attribute data points, we introduce an adjustable hyper-parameter $\beta$ that balances reconstruction accuracy between edges and attributes. In addition, similar to [19], we wish the parameters of our predictive distribution, i.e. $q_{\boldsymbol{\phi}^c}(\mathbf{Y}_v \mid \phi(\mathbf{F}_v^n))$, can also be trained within the labelled nodes based on their feature $\mathbf{F}_v^n$; therefore, we add a classification loss to Eq. 8 and introduce a hyper-parameter $\alpha$ to govern the relative weight between generative and purely discriminative models, which results in the following loss:

$$\mathcal{J}(\mathbf{A}, \mathbf{X}, \mathbf{Y}^l) = \beta \cdot \Big( \sum_{i,j \in \mathcal{V}^l} \mathcal{L}(\mathcal{O}_{ij}^{ll}) + \sum_{i \in \mathcal{V}^u, j \in \mathcal{V}^u} \mathcal{U}(\mathcal{O}_{ij}^{uu}) + \sum_{i \in \mathcal{V}^l, j \in \mathcal{V}^u} \mathcal{M}(\mathcal{O}_{ij}^{lu}) \Big)$$
$$+ (1 - \beta) \cdot \Big( \sum_{i \in \mathcal{V}^l, a \in \mathcal{A}} \mathcal{B}(\mathcal{O}_{ia}^{la}) + \sum_{i \in \mathcal{V}^u, a \in \mathcal{A}} \mathcal{C}(\mathcal{O}_{ia}^{ua}) \Big) + \alpha \cdot \mathbb{E}_{v \sim \mathcal{V}^l}[- \log q_{\boldsymbol{\phi}^c}(\mathbf{Y}_v \mid \phi(\mathbf{F}_v^n))]. \tag{9}$$

**Optimization.** The parameters, i.e. $\boldsymbol{\theta} = (\boldsymbol{\theta}^e, \boldsymbol{\theta}^a)$ and $\boldsymbol{\phi} = (\boldsymbol{\phi}^n, \boldsymbol{\phi}^a, \boldsymbol{\phi}^c)$ (Fig. 2), of the generative and inference networks are jointly trained by optimizing Eq. 9 using gradient descent[2].

We assume the priors and the variational posteriors of $\mathbf{Z}^n$ and $\mathbf{Z}^a$ to be Gaussian distributions, then the KL-divergence terms in Eq. 9 have analytical forms. However, analytical solutions of expectations w.r.t. these two variational posteriors are still intractable in the general case. To address this problem, we can reduce the problem of estimating the gradient w.r.t. parameters of the posterior distribution to the simpler problem of estimating the gradient w.r.t. parameters of a deterministic function, which is called the *reparameterization* trick [18, 32]. Specifically, we sample noise $\boldsymbol{\epsilon} \sim \mathcal{N}(0, \mathbf{I}_D)$ and reparameterize $\mathbf{Z}_i^n = \boldsymbol{\mu}_{\boldsymbol{\phi}^n} + \boldsymbol{\epsilon}\boldsymbol{\sigma}_{\boldsymbol{\phi}^n}$ (or $\mathbf{Z}_i^a = \boldsymbol{\mu}_{\boldsymbol{\phi}^a} + \boldsymbol{\epsilon}\boldsymbol{\sigma}_{\boldsymbol{\phi}^a}$ for attributes). By doing so, the value of $\mathbf{Z}_i^n$ (or $\mathbf{Z}_i^a$) is deterministic given both the parameters of variational posterior distributions, so that the stochasticity in the sampling process is isolated and the gradient with respect to $\boldsymbol{\mu}_{\boldsymbol{\phi}^n}$ and $\boldsymbol{\sigma}_{\boldsymbol{\phi}^n}$ (or $\boldsymbol{\mu}_{\boldsymbol{\phi}^a}$ and $\boldsymbol{\sigma}_{\boldsymbol{\phi}^a}$) can be back-propagated through the sampled $\mathbf{Z}_i^n$ (or $\mathbf{Z}_i^a$).

However, this trick is unavailable for optimizing the parameters of the latent label distribution $q_{\boldsymbol{\phi}^c}(\mathbf{Y}_i^u \mid \mathbf{F}_i^n)$, because the categorical distribution is not reparameterizable. Kingma et.al [19] approach this by marginalizing out $\mathbf{Y}_i^u$ over all classes, so that for unlabelled data, inference is still on $q_{\boldsymbol{\phi}^c}(\mathbf{Y}_i^u \mid \mathbf{F}_i^n)$ for each $\mathbf{Y}_i^u$. As we have mentioned in subsection 3.1, this simple solution is prohibitively expensive for a large number of classes, especially in our case where we have double expectation over $q_{\boldsymbol{\phi}^c}(\mathbf{Y}_i^u \mid \mathbf{F}_i^n)$ and $q_{\boldsymbol{\phi}^c}(\mathbf{Y}_j^u \mid \mathbf{F}_j^n)$ (factorized from $q_{\boldsymbol{\phi}^c}(\mathbf{Y}_i^u, \mathbf{Y}_j^u \mid \mathbf{F}_i^n, \mathbf{F}_j^n)$ of $\mathcal{U}(\mathcal{O}_{ij}^{uu})$, please refer to Eq. 18 in the supplementary material). In this paper, we alleviate this by applying the Gumbel-Softmax trick.

The Gumbel-Softmax trick [33, 34] provides a continuous and differentiable approximation to draw categorical samples $\mathbf{Y}_i^u$ from a categorical distribution with class probabilities $\boldsymbol{\pi}_{\boldsymbol{\phi}^c}$:

$$\mathbf{Y}_{ik}^u = \frac{\exp(\log \boldsymbol{\pi}_{\boldsymbol{\phi}^c,k} + g_k)/\tau}{\sum_{k=1}^K \exp(\log \boldsymbol{\pi}_{\boldsymbol{\phi}^c,k} + g_k)/\tau} , \tag{10}$$

where $\{g_k\}_{k=1}^K$ are i.i.d. samples drawn from the $\mathrm{Gumbel}(0, 1)$ distribution, and $\tau$ is the softmax temperature which is set to be $0.2$ in our experiments. Samples from the Gumbel-Softmax distribution become one-hot when $\tau \to 0$ and smooth when $\tau > 0$. With this trick, Gumbel-Softmax allows us to backpropagate through $\mathbf{Y}_i^u \sim q_{\boldsymbol{\phi}^c}(\mathbf{Y}^u \mid \mathbf{F}^n)$ for single sample gradient estimation.

# 5 Experiments

The research questions guiding the remainder of this paper are: (**RQ**1) How is performance of our SCAN in semi-supervised node classification task? (**RQ**2) Can our SCAN perform better than other models in the attribute inference task, where capturing the affinities between nodes and attributes is crucial? (**RQ**3) Can our SCAN learn meaningful embeddings for the task of network visualizations?

## 5.1 Experimental Settings

To evaluate the performance of our SCAN model on semi-supervised node classification task, four state-of-the-art semi-supervised network embedding methods are included for comparisons: **Planetoid-T** [11], **GCN** [12], **SEANO** [13] and **GraphSGAN** [14]. We also evaluate the learned embeddings of SCAN by the attribute inference task, comparing to four attribute inference baselines: **SAN** [16], **EdgeExp** [35], **BLA** [15] and **CAN** [24]. All experiments of this paper are conducted base on three real-world attributed networks, i.e. **Pubmed** [12], **BlogCatalog** [36] and **Flickr** [36].

For our SCAN model, the inferred embeddings of nodes can be directly used as the input features of a classifier to predict the class labels. Here we apply Support Vector Machine (SVM) as our classifier, which we refer to it as **SCAN_SVM**. In addition, the inference model of SCAN (i.e. the *discriminative network*) also infers the latent labels for all the unlabelled nodes during the inference process, which can be used as another classifier, namely the **SCAN_DIS**. [1]

Table 1: Performance of semi-supervised node classification. The best and the second best performance runs per metric per dataset are marked in boldface and underlined, respectively.

| Method | Pubmed | | | Flickr | | | BlogCatalog | | |
|---|---|---|---|---|---|---|---|---|---|
| | Ma_F1 | Mi_F1 | ACC | Ma_F1 | Mi_F1 | ACC | Ma_F1 | Mi_F1 | ACC |
| **SEANO** | .841 | .845 | .850 | .738 | .748 | .741 | .627 | .635 | .648 |
| **Planetoid-T** | .815 | .825 | .823 | .721 | .743 | .733 | .803 | .811 | .817 |
| **GCN** | .838 | .847 | .850 | .286 | .291 | .309 | .509 | .527 | .538 |
| **GraphSGAN** | .839 | .842 | .841 | .697 | .715 | .702 | .698 | .703 | .719 |
| **SCAN_SVM** | $\underline{.847}^{\dagger}$ | $\underline{.852}^{\dagger}$ | $\underline{.851}$ | $\underline{.747}^{\dagger}$ | $\underline{.750}$ | $\underline{.747}^{\dagger}$ | $\underline{.820}^{\dagger}$ | $\underline{.829}^{\dagger}$ | $\underline{.835}^{\dagger}$ |
| **SCAN_DIS** | $\mathbf{.850}^{\dagger}$ | $\mathbf{.858}^{\dagger}$ | $\mathbf{.862}^{\dagger}$ | $\mathbf{.749}^{\dagger}$ | $\mathbf{.753}^{\dagger}$ | $\mathbf{.750}^{\dagger}$ | $\mathbf{.829}^{\dagger}$ | $\mathbf{.832}^{\dagger}$ | $\mathbf{.839}^{\dagger}$ |

## 5.2 Results

**Semi-supervised Node Classification.** To answer (**RQ1**), we conduct semi-supervised node classification experiments in the three networks. Specifically, we randomly select 10% of nodes as the labelled nodes, train our SCAN model to obtain the embeddings of nodes and predict the labels for the unlabelled nodes by the discriminator, and then feed the obtained embeddings of nodes into an SVM classifier to predict the labels. We repeatedly this process 10 times and report the average performance. As for evaluation metrics, we employ macro F1 (Ma_F1), micro F1 (Mi_F1) and accuracy (ACC) to measure the performance of semi-supervised node classification. Tab. 1 shows the result of our SCAN methods and other four baseline methods on our datasets[2]. As shown in the table, both our SCAN_DIS and SCAN_SVM can outperform the baseline models, and SCAN_DIS always achieves the best performance in all the metrics with significant improvement. It is worth noting that while some baselines like SEANO and GCN perform poorly on BlogCatalog dataset, our methods, both SCAN_DIS and SCAN_SVM, can still achieve significantly better performance. This result shows that our model is not only capable of accurately classifying the nodes by the discriminative network, but also capable of learning effective representations of nodes for the attributed networks.

**Attribute Inference.** Subsequently, we answer **RQ3** and aim at understanding the performance of SCAN and the baselines over attribute inference task. Attribute inference aims at predicting the value of attributes of the nodes. In this task, we take four state-of-the-art attribute inference algorithms,

namely SAN [16], EdgeExp [35] BLA [15] and CAN [24], as our baselines for performance comparison. We adopt the same experimental setting as in [37, 12], we randomly divide all edges into three sets, i.e., the training (85%), validating (5%) and testing (10%) sets, and employ area under the ROC curve (AUC) and average precision (AP) scores as evaluation metrics to measure the attribute inference performance. Tab. 2 presents the attribute inference performance of our SCAN and the baseline models in the three attributed networks. We can observe that our model outperforms all the baseline models in all the datasets, and the improvement is significant in Pubmed and BlogCatalog networks. This can be explained by the fact that our SCAN optimizes a loss function consisting of the reconstruction error of all the attributes. This again shows that our co-embedding model can learn effective representations for both nodes and attributes where the infinities between nodes and attributes can be effectively captured and measured.

| Method | Pubmed | | Flickr | | BlogCatalog | |
|---|---|---|---|---|---|---|
| | AUC | AP | AUC | AP | AUC | AP |
| **EdgeExp** | .586 | .576 | .678 | .685 | .684 | .744 |
| **SAN** | .579 | .572 | .653 | .660 | .694 | .710 |
| **BLA** | .622 | .602 | .730 | .769 | .787 | .792 |
| **CAN** | .670 | .652 | .867 | .868 | .867 | .865 |
| **SCAN** | **.713**$^{\dagger}$ | **.682**$^{\dagger}$ | **.874**$^{\dagger}$ | **.871** | **.893**$^{\dagger}$ | **.895**$^{\dagger}$ |

Table 2: Attribute inference performance. The best runs per metric per dataset are marked in boldface.

**Network Visualization.** Finally, to further evaluate the qualities of embeddings in our approach, we make a comparison on the visualization of node representation in Fig. 3. Specifically, we obtain all the 64-dimensional embeddings of nodes for each comparison methods, then use the t-SNE tool [38] to transfer them into 2-D vectors and plot these vectors on 2-D planes. We can find that our approach can achieve more compact and separated clusters compared with the baseline methods. This result can also explain why our approach achieves better performance on node classification task. We also visualize the variances of each node representation by **ellipsoids**, where the 2-D variances are obtained by dividing all the dimension into two groups and then calculating average variances of them. We see that some of these nodes have large variances as their features are relatively sparser, resulting in more uncertainty of their representations.

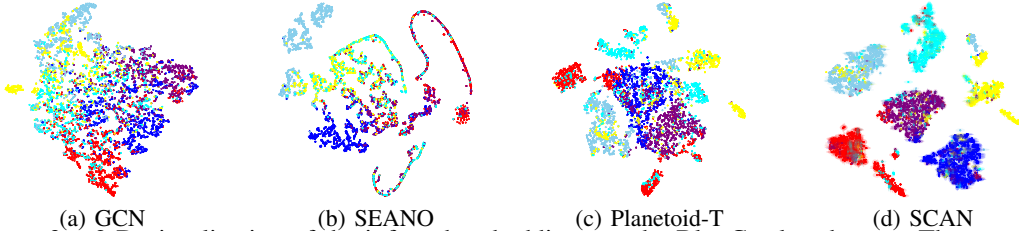

| (a) GCN | (b) SEANO | (c) Planetoid-T | (d) SCAN |

Figure 3: 2-D visualization of the inferred embeddings on the BlogCatalog dataset. The same colour indicates the same class label. Ellipsoids surrounding the nodes in our model are the averaged variances indicating the uncertainty of their embeddings. Note that the ellipsoids of our model need to be zoomed in to be visible.

## 6    Conclusion

We aim at solving the problem of embedding attributed networks in a semi-supervised way. We showed how the SVAE can be generalized to the heterogeneous data, and proposed a semi-supervised co-embedding model (called SCAN) to solve the problem. Our SCAN learns low-dimensional Gaussian embeddings for both nodes and attributes in the same semantic space in a semi-supervised way, such that the affinities between nodes and their attributes and the similarities among nodes can be effectively measured with the uncertainty can be preserved. Meanwhile, it is also able to learn an effective discriminative model to generalize from small labelled data to large unlabelled ones. Our experiments showed that our SCAN model can yield excellent and better performance compared with the state-of-the-art baselines in various applications, including semi-supervised node classification and attribute inference, and leverage the expressive power to obtain high-quality representations of both nodes and attributes. As to future work, we intend to extend our SCAN model to heterogeneous networks, and embed nodes and attributes for dynamic attributed networks.

**Acknowledgments.** This research was partially supported by the National Natural Science Foundation of China (Grant No. 61906219).

## Footnotes

[1]The code of our SCAN is publicly available from: `https://github.com/mengzaiqiao/SCAN`.

[1]The detail derivations of the five ELBOs can be found in the supplementary material.

[2]The implementation details of the inference and generative networks are given in our supplementary material.

[1]Other experimental setting details and parameter sensitivity analysis of our model can be found in supplementary material, due to the space limitation.

[2]In each dataset, significant improvements over the comparative methods (other than ours) are marked with $\dagger$ (paired t-test, $p < .05$)

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
