[Supplementary Material]

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

# A Appendix

## A.1 Main Notations

The following Tab. 3 lists the notations used throughout this paper.

Table 3: Main notations used in the paper.

| Symble | Desription |
|---|---|
| $\mathcal{G}$ | partially labelled attributed network |
| $\mathcal{V}$ | set of nodes |
| $\mathcal{V}^l$ | set of labelled nodes |
| $\mathcal{V}^u$ | set of unlabelled nodes |
| $\mathcal{A}$ | set of attributes |
| $N = |\mathcal{V}|$ | number of nodes |
| $M = |\mathcal{A}|$ | number of attributes |
| $K$ | number of the class labels |
| $D$ | dimension of latent variables |
| $\mathbf{A} \in \mathbb{R}^{N \times N}$ | *adjacency* matrix |
| $\mathbf{X} \in \mathbb{R}^{N \times M}$ | node *attribute* matrix |
| $\mathbf{Y}^l \in \{0,1\}^{|\mathcal{V}^l| \times K}$ | label matrix of labelled nodes |
| $\mathbf{Y}^u \in \{0,1\}^{|\mathcal{V}^u| \times K}$ | label matrix of unlabelled nodes |
| $\mathbf{F}^n \in \mathbb{R}^{N \times (N+M)}$ | features of nodes |
| $\mathbf{F}^a \in \mathbb{R}^{M \times N}$ | features of attributes |
| $\mathbf{Z}^n \in \mathbb{R}^{N \times D}$ | latent representation matrix for all *nodes* |
| $\mathbf{Z}^a \in \mathbb{R}^{M \times D}$ | latent representation matrix for all *attributes* |
| $\boldsymbol{\theta}^e$ | parameters of the generative network for edges |
| $\boldsymbol{\theta}^n$ | parameters of the generative network for attributes |
| $\boldsymbol{\phi}^n$ | parameters of the inference network for nodes |
| $\boldsymbol{\phi}^a$ | parameters of the inference network for attributes |
| $\boldsymbol{\phi}^c$ | parameters of the discriminative network for labels |

## A.2 Derivations of the ELBOs

**Case 1: Edges connecting two labelled nodes**

To simplify notations, we let $\mathcal{O}_{ij}^{ll} = (\mathbf{A}_{ij}, \mathbf{Y}_i^l, \mathbf{Y}_j^l)$ be the observed variables of an edge connecting two labelled nodes, where $\mathbf{A}_{ij} \in \mathbf{A}$ is the edge weight and $\mathbf{Y}_i^l$ and $\mathbf{Y}_j^l$ are labels of the two nodes, $\mathcal{Z}_{ij}^{ll} = (\mathbf{Z}_i^n, \mathbf{Z}_j^n)$ be the collection of latent variables of the two labelled nodes, and $\mathcal{F}_{ij}^{ll} = (\mathbf{F}_i^n, \mathbf{F}_j^n, \mathbf{Y}_i^l, \mathbf{Y}_j^l)$ be the conditional variables of the variational posterior, where $\mathbf{F}_i^n = [\mathbf{A}_i; \mathbf{X}_i]$ and $\mathbf{F}_j^n = [\mathbf{A}_j; \mathbf{X}_j]$ are the input features of nodes $i$ and $j$, respectively, and $[\cdot; \cdot]$ is the concatenation operator. We introduce a variational distribution over latent variables $\mathcal{Z}_{ij}^{ll}$ conditioned on $\mathcal{F}_{ij}^{ll}$, and using the Jensen's inequality, we can obtain the following inequality:

$$\log p(\mathcal{O}_{ij}^{ll}) \geq \mathbb{E}_{q_\phi(\mathcal{Z}_{ij}^{ll}|\mathcal{F}_{ij}^{ll})}[\log p_\theta(\mathcal{O}_{ij}^{ll}, \mathcal{Z}_{ij}^{ll}) - \log q_\phi(\mathcal{Z}_{ij}^{ll} \mid \mathcal{F}_{ij}^{ll})]. \tag{11}$$

Here $q_\phi(\mathcal{Z}_{ij}^{ll} \mid \mathcal{F}_{ij}^{ll})$ is the variational posterior parameterized by $\phi$ of the inference network, and $p_\theta(\mathcal{O}_{ij}^{ll}, \mathcal{Z}_{ij}^{ll})$ is the joint distribution of both the observed and latent variables, which can be written as:

$$\begin{aligned} p_\theta(\mathcal{O}_{ij}^{ll}, \mathcal{Z}_{ij}^{ll}) =& p_\theta(\mathbf{A}_{ij}, \mathbf{Y}_i^l, \mathbf{Y}_j^l, \mathbf{Z}_i^n, \mathbf{Z}_j^n) \\ =& p(\mathbf{Z}_i^n)p(\mathbf{Z}_j^n)p(\mathbf{Y}_i^l)p(\mathbf{Y}_j^l)p_{\theta^e}(\mathbf{A}_{ij} \mid \mathbf{Z}_i^n, \mathbf{Z}_j^n, \mathbf{Y}_i^l, \mathbf{Y}_j^l), \end{aligned} \tag{12}$$

where $\theta^e$ is the parameter of the **generative model for edges**. Since the true posterior $p(\mathcal{Z}_{ij}^{ll} \mid \mathcal{O}_{ij}^{ll})$ is intractable, we introduce the variational posterior, i.e. $q_\phi(\mathcal{Z}_{ij}^{ll} \mid \mathcal{F}_{ij}^{ll})$, to approximate the true

posterior, and it can be factorized as the following:

$$q_{\phi}(\mathcal{Z}_{ij}^{ll} \mid \mathcal{F}_{ij}^{ll}) = q_{\phi}(\mathbf{Z}_i^n, \mathbf{Z}_j^n \mid \mathbf{F}_i^n, \mathbf{F}_j^n, \mathbf{Y}_i^l, \mathbf{Y}_j^l)$$

$$= q_{\phi^n}(\mathbf{Z}_i^n \mid \mathbf{F}_i^n, \mathbf{Y}_i^l) q_{\phi^n}(\mathbf{Z}_j^n \mid \mathbf{F}_j^n, \mathbf{Y}_j^l), \tag{13}$$

where $\phi^n$ is the parameter of the **inference model for nodes**. Substituting Eq. 12 and Eq. 13 into Eq. 11, Eq. 11 can be represented as:

$$\log p(\mathcal{O}_{ij}^{ll}) \geq \mathbb{E}_{q_{\phi}(\mathcal{Z}_{ij}^{ll} \mid \mathcal{F}_{ij}^{ll})} [\log p_{\boldsymbol{\theta}^e}(\mathbf{A}_{ij} \mid \mathbf{Z}_i^n, \mathbf{Z}_j^n, \mathbf{Y}_i^l, \mathbf{Y}_j^l) + \log p(\mathbf{Y}_i^l) p(\mathbf{Y}_j^l)]$$

$$- KL(q_{\phi^n}(\mathbf{Z}_i^n \mid \mathbf{F}_i^n, \mathbf{Y}_i^l) \mid p(\mathbf{Z}_i^n)) - KL(q_{\phi^n}(\mathbf{Z}_j^n \mid \mathbf{F}_j^n, \mathbf{Y}_j^l) \mid p(\mathbf{Z}_j^n))$$

$$\triangleq - \mathcal{L}(\mathcal{O}_{ij}^{ll}), \tag{14}$$

where $\mathcal{L}(\mathcal{O}_{ij}^{ll})$ is denoted as the ELBO on the marginal likelihood of $\mathcal{O}_{ij}^{ll}$, and $KL(\cdot \mid \cdot)$ is the Kullback-Leibler (KL) divergence.

**Case 2: Edges connecting two unlabelled nodes**

In this case, labels of the two nodes are treated as latent variables over which we perform variational posterior inference to infer their missing labels. Similarly, we let $\mathcal{O}_{ij}^{uu} = (\mathbf{A}_{ij})$ be the observed variable for edge weight between nodes $i$ and $j$, $\mathcal{Z}_{ij}^{uu} = (\mathbf{Z}_i^n, \mathbf{Z}_j^n, \mathbf{Y}_i^u, \mathbf{Y}_j^u)$ be the collection of latent variables of the two nodes, and $\mathcal{F}_{ij}^{uu} = (\mathbf{F}_i^n, \mathbf{F}_j^n)$ be conditional variables of the inference model. Then, the logarithm marginal likelihood of $\mathcal{O}_{ij}^{uu}$ can be written as:

$$\log p(\mathcal{O}_{ij}^{uu}) \geq \mathbb{E}_{q_{\phi^n}(\mathcal{Z}_{ij}^{uu} \mid \mathcal{F}_{ij}^{uu})} [\log p_{\boldsymbol{\theta}}(\mathcal{O}_{ij}^{uu}, \mathcal{Z}_{ij}^{uu}) - \log q_{\phi^n}(\mathcal{Z}_{ij}^{uu} \mid \mathcal{F}_{ij}^{uu})], \tag{15}$$

where the joint distribution of both the observed and latent variables, i.e. $p_{\boldsymbol{\theta}}(\mathcal{O}_{ij}^{uu}, \mathcal{Z}_{ij}^{uu})$, can be represented as:

$$p_{\boldsymbol{\theta}}(\mathcal{O}_{ij}^{uu}, \mathcal{Z}_{ij}^{uu}) = p_{\boldsymbol{\theta}}(\mathbf{A}_{ij}, \mathbf{Y}_i^u, \mathbf{Y}_j^u, \mathbf{Z}_i^n, \mathbf{Z}_j^n)$$

$$= p(\mathbf{Z}_i^n) p(\mathbf{Z}_j^n) p(\mathbf{Y}_i^u) p(\mathbf{Y}_j^u) p_{\boldsymbol{\theta}^e}(\mathbf{A}_{ij} \mid \mathbf{Z}_i^n, \mathbf{Z}_j^n, \mathbf{Y}_i^u, \mathbf{Y}_j^u), \tag{16}$$

and the variational posterior $q_{\phi}(\mathcal{Z}_{ij}^{uu} \mid \mathcal{F}_{ij}^{uu})$ can be factorized as:

$$q_{\phi}(\mathcal{Z}_{ij}^{uu} \mid \mathcal{F}_{ij}^{uu}) = q_{\phi}(\mathbf{Z}_i^n, \mathbf{Z}_j^n, \mathbf{Y}_i^u, \mathbf{Y}_j^u \mid \mathbf{F}_i^n, \mathbf{F}_j^n)$$

$$= q_{\phi^n}(\mathbf{Z}_i^n \mid \mathbf{F}_i^n, \mathbf{Y}_i^u) q_{\phi^n}(\mathbf{Z}_j^n \mid \mathbf{F}_j^n, \mathbf{Y}_j^u) q_{\phi^c}(\mathbf{Y}_i^u \mid \mathbf{F}_i^n) q_{\phi^c}(\mathbf{Y}_j^u \mid \mathbf{F}_j^n), \tag{17}$$

where $\phi^c$ is the parameters of the **inference model for nodes' class labels (discriminative network)**. Substituting Eq. 16 and Eq. 17 into Eq. 15, Eq. 15 can be represented as:

$$\log p(\mathcal{O}_{ij}^{uu}) \geq \mathbb{E}_{q_{\phi}(\mathcal{Z}_{ij}^{uu} \mid \mathcal{F}_{ij}^{uu})} [\log p_{\boldsymbol{\theta}^e}(\mathbf{A}_{ij} \mid \mathbf{Z}_i^n, \mathbf{Z}_j^n, \mathbf{Y}_i^u, \mathbf{Y}_j^u) + \log p(\mathbf{Z}_i^n) + \log p(\mathbf{Z}_j^n)$$

$$+ \log p(\mathbf{Y}_i^u) + \log p(\mathbf{Y}_j^u) - \log q_{\phi^n}(\mathbf{Z}_i^n \mid \mathbf{F}_i^n, \mathbf{Y}_i^u) - \log q_{\phi^n}(\mathbf{Z}_j^n \mid \mathbf{F}_j^n, \mathbf{Y}_j^u)$$

$$- \log q_{\phi^c}(\mathbf{Y}_i^u \mid \mathbf{F}_i^n) - \log q_{\phi^c}(\mathbf{Y}_j^u \mid \mathbf{F}_j^n)]$$

$$= \mathbb{E}_{q_{\phi^c}(\mathbf{Y}_i^u, \mathbf{Y}_j^u \mid \mathbf{F}_i^n, \mathbf{F}_j^n)} [-\mathcal{L}(\mathbf{A}_{ij})] + \mathcal{H}(q_{\phi^c}(\mathbf{Y}_i^u \mid \mathbf{F}_i^n)) + \mathcal{H}(q_{\phi^c}(\mathbf{Y}_j^u \mid \mathbf{F}_j^n))$$

$$\triangleq -\mathcal{U}(\mathcal{O}_{ij}^{uu}) \tag{18}$$

where $\mathcal{U}(\mathcal{O}_{ij}^{uu})$ is denoted as the ELBO of $\mathcal{O}_{ij}^{uu}$, $\mathcal{H}(q_{\phi^c}(\mathbf{Y}_i^u \mid \mathbf{F}_i^n))$ and $\mathcal{H}(q_{\phi^c}(\mathbf{Y}_j^u \mid \mathbf{F}_j^n))$ are the entropies of the label variational distributions $q_{\phi^c}(\mathbf{Y}_i^u \mid \mathbf{F}_i^n)$ and $q_{\phi^c}(\mathbf{Y}_j^u \mid \mathbf{F}_j^n)$, respectively.

**Case 3: Edges connecting an unlabelled node and a labelled node**

Let $\mathcal{O}_{ij}^{lu} = (\mathbf{A}_{ij}, \mathbf{Y}_i^l)$ be the observation collection of an edge connecting an unlabelled node $i$ and a labelled node $j$, $\mathcal{Z}_{ij}^{lu} = (\mathbf{Z}_i^n, \mathbf{Z}_j^n, \mathbf{Y}_j^u)$ be the collection of latent variables of the two nodes, and $\mathcal{F}_{ij}^{lu} = (\mathbf{F}_i^n, \mathbf{F}_j^n, \mathbf{Y}_i^l)$ be the conditional variables of variational posterior. Then, the logarithm marginal likelihood of $\mathcal{O}_{ij}^{lu}$ can be written as:

$$\log p(\mathcal{O}_{ij}^{lu}) \geq \mathbb{E}_{q_{\phi}(\mathcal{Z}_{ij}^{lu} \mid \mathcal{F}_{ij}^{lu})} [\log p_{\boldsymbol{\theta}}(\mathcal{O}_{ij}^{lu}, \mathcal{Z}_{ij}^{lu}) - \log q_{\phi}(\mathcal{Z}_{ij}^{lu} \mid \mathcal{F}_{ij}^{lu})], \tag{19}$$

where the joint distribution of both the observed and latent variables, i.e. $p_{\boldsymbol{\theta}}(\mathcal{O}_{ij}^{lu}, \mathcal{Z}_{ij}^{lu})$, can be represented as:

$$
\begin{aligned}
p_{\boldsymbol{\theta}}(\mathcal{O}_{ij}^{lu}, \mathcal{Z}_{ij}^{lu}) &= p_{\boldsymbol{\theta}}(\mathbf{A}_{ij}, \mathbf{Z}_i^n, \mathbf{Z}_j^n, \mathbf{Y}_i^l, \mathbf{Y}_j^u) \\
&= p(\mathbf{Z}_i^n)p(\mathbf{Z}_j^n)p(\mathbf{Y}_i^l)p(\mathbf{Y}_j^u)p_{\boldsymbol{\theta}^e}(\mathbf{A}_{ij} \mid \mathbf{Z}_i^n, \mathbf{Z}_j^n, \mathbf{Y}_i^l, \mathbf{Y}_j^u),
\end{aligned} \tag{20}
$$

and the variational posterior $q_{\boldsymbol{\phi}}(\mathcal{Z}_{ij}^{lu} \mid \mathcal{F}_{ij}^{lu})$ can be factorized as:

$$
\begin{aligned}
q_{\boldsymbol{\phi}}(\mathcal{Z}_{ij}^{lu} \mid \mathcal{F}_{ij}^{lu}) &= q_{\boldsymbol{\phi}}(\mathbf{Z}_i^n, \mathbf{Z}_j^n, \mathbf{Y}_j^u \mid \mathbf{F}_i^n, \mathbf{F}_j^n, \mathbf{Y}_i^l) \\
&= q_{\boldsymbol{\phi}^n}(\mathbf{Z}_i^n \mid \mathbf{F}_i^n, \mathbf{Y}_i^l)q_{\boldsymbol{\phi}^n}(\mathbf{Z}_j^n \mid \mathbf{F}_j^n, \mathbf{Y}_j^u)q_{\boldsymbol{\phi}^c}(\mathbf{Y}_j^u \mid \mathbf{F}_j^n).
\end{aligned} \tag{21}
$$

Substituting Eq. 20 and Eq. 21 into Eq. 19, Eq. 19 can be written as:

$$
\begin{aligned}
\log p(\mathcal{O}_{ij}^{lu}) \geq\ & \mathbb{E}_{q_{\boldsymbol{\phi}}(\mathcal{Z}_{ij}^{lu}\mid\mathcal{F}_{ij}^{lu})}[\log p_{\boldsymbol{\theta}^e}(\mathbf{A}_{ij} \mid \mathbf{Z}_i^n, \mathbf{Z}_j^n, \mathbf{Y}_i^l, \mathbf{Y}_j^u) \\
& + \log p(\mathbf{Z}_i^n)p(\mathbf{Z}_j^n) + \log p(\mathbf{Y}_i^l)p(\mathbf{Y}_j^u) - \log q_{\boldsymbol{\phi}^n}(\mathbf{Z}_i^n \mid \mathbf{F}_i^n, \mathbf{Y}_i^u) \\
& - \log q_{\boldsymbol{\phi}^n}(\mathbf{Z}_j^n \mid \mathbf{F}_j^n, \mathbf{Y}_j^u) - \log q_{\boldsymbol{\phi}^c}(\mathbf{Y}_j^u \mid \mathbf{F}_j^n)] \\
=\ & \mathbb{E}_{q_{\boldsymbol{\phi}^c}(\mathbf{Y}_j^u\mid\mathbf{F}_j^n)}[-\mathcal{L}(\mathbf{A}_{ij})] + \mathcal{H}(q_{\boldsymbol{\phi}^c}(\mathbf{Y}_j^u \mid \mathbf{F}_j^n)) \\
\triangleq\ & -\mathcal{M}(\mathcal{O}_{ij}^{lu}),
\end{aligned} \tag{22}
$$

where $\mathcal{M}(\mathcal{O}_{ij}^{lu})$ is denoted as the ELBO of $\mathcal{O}_{ij}^{lu}$.

## Case 4: Attributes associated with the labelled nodes

In this case, we let $\mathcal{O}_{ia}^{la} = (\mathbf{X}_{ia}, \mathbf{Y}_i^l)$ be the observed variables, $\mathcal{Z}_{ia}^{la} = (\mathbf{Z}_i^n, \mathbf{Z}_a^a)$ be the collection of latent variables of node $i$ and attribute $a$, and $\mathcal{F}_{ia}^{la} = (\mathbf{F}_i^n, \mathbf{F}_a^a, \mathbf{Y}_i^l)$ be the conditional variables on variational posteriors, where $\mathbf{F}_a^a = \mathbf{X}_a^T$ is the input feature of attribute $a$. Then the logarithm marginal likelihood of $\mathcal{O}_{ia}^{la}$ can be written as:

$$
\log p(\mathcal{O}_{ia}^{la}) \geq \mathbb{E}_{q_{\boldsymbol{\phi}}(\mathcal{Z}_{ia}^{la}\mid\mathcal{F}_{ia}^{la})}[\log p_{\boldsymbol{\theta}}(\mathcal{O}_{ia}^{la}, \mathcal{Z}_{ia}^{la}) - \log q_{\boldsymbol{\phi}}(\mathcal{Z}_{ia}^{la} \mid \mathcal{F}_{ia}^{la})], \tag{23}
$$

where the joint distribution of both the observed and latent variables, i.e. $p_{\boldsymbol{\theta}}(\mathcal{O}_{ia}^{la}, \mathcal{Z}_{ij}^{la})$, can be represented as:

$$
\begin{aligned}
\log p_{\boldsymbol{\theta}}(\mathcal{O}_{ia}^{la}, \mathcal{Z}_{ia}^{la}) &= p_{\boldsymbol{\theta}}(\mathbf{X}_{ia}, \mathbf{Y}_i^l, \mathbf{Z}_i^n, \mathbf{Z}_a^a) \\
&= p(\mathbf{Z}_i^n)p(\mathbf{Z}_a^a)p(\mathbf{Y}_i^l)p_{\boldsymbol{\theta}^a}(\mathbf{X}_{ia} \mid \mathbf{Z}_i^n, \mathbf{Z}_a^a, \mathbf{Y}_i^l),
\end{aligned} \tag{24}
$$

where $\boldsymbol{\theta}^a$ is the parameter of the **generative model for attributes**. The variational posterior $q_{\boldsymbol{\phi}}(\mathcal{Z}_{ia}^{la})$ can be factorized as:

$$
\begin{aligned}
q_{\boldsymbol{\phi}}(\mathcal{Z}_{ia}^{la} \mid \mathcal{F}_{ia}^{la}) &= q_{\boldsymbol{\phi}}(\mathbf{Z}_i^n, \mathbf{Z}_a^a \mid \mathbf{F}_i^n, \mathbf{F}_a^a, \mathbf{Y}_i^l) \\
&= q_{\boldsymbol{\phi}^n}(\mathbf{Z}_i^n \mid \mathbf{F}_i^n, \mathbf{Y}_i^l)q_{\boldsymbol{\phi}^a}(\mathbf{Z}_a^a \mid \mathbf{F}_a^a),
\end{aligned} \tag{25}
$$

where $\mathbf{F}_a^a$ is the input feature of attribute $a$ and $\phi^a$ is the parameter of the **inference model for attributes**. Substituting Eq. 24 and Eq. 25 into Eq. 23, Eq. 23 can be represented as:

$$
\begin{aligned}
\log p(\mathcal{O}_{ia}^{la}) \geq\ & \mathbb{E}_{q_{\boldsymbol{\phi}}(\mathcal{Z}_{ia}^{la}\mid\mathcal{F}_{ia}^{la})}[\log p_{\boldsymbol{\theta}^a}(\mathbf{X}_{ia} \mid \mathbf{Z}_i^n, \mathbf{Z}_a^a, \mathbf{Y}_i^l) + \log p(\mathbf{Y}_i^l)] \\
& - KL(q_{\boldsymbol{\phi}^n}(\mathbf{Z}_i^n \mid \mathbf{F}_i^n, \mathbf{Y}_i^l) \mid p(\mathbf{Z}_i^n)) - KL(q_{\boldsymbol{\phi}^a}(\mathbf{Z}_a^a \mid \mathbf{F}_a^a) \mid p(\mathbf{Z}_a^a))] \\
\triangleq\ & -\mathcal{B}(\mathcal{O}_{ia}^{la}),
\end{aligned} \tag{26}
$$

where $\mathcal{B}(\mathcal{O}_{ia}^{la})$ is denoted as the ELBO of $\mathcal{O}_{ia}^{la}$.

## Case 5: Attributes associated with the unlabelled nodes

We let $\mathcal{O}_{ia}^{ua} = (\mathbf{X}_{ia})$ be an observed attribute value, $\mathcal{Z}_{ia}^{ua} = (\mathbf{Z}_i^n, \mathbf{Z}_a^a, \mathbf{Y}_i^u)$ be the collection of latent variables of node $i$ and attribute $a$, and $\mathcal{F}_{ia}^{ua} = (\mathbf{F}_i^n, \mathbf{F}_a^a)$ be the conditional variables. Then the logarithm marginal likelihood of $\mathcal{O}_{ij}^{ua}$ can be written as:

$$
\log p(\mathcal{O}_{ia}^{ua}) \geq \mathbb{E}_{q_{\boldsymbol{\phi}}(\mathcal{Z}_{ia}^{ua}\mid\mathcal{F}_{ia}^{ua})}[\log p_{\boldsymbol{\theta}}(\mathcal{O}_{ia}^{ua}, \mathcal{Z}_{ia}^{ua}) - \log q_{\boldsymbol{\phi}}(\mathcal{Z}_{ia}^{ua} \mid \mathcal{F}_{ia}^{ua})], \tag{27}
$$

where the joint distribution of both the observed and latent variables, i.e. $p_{\boldsymbol{\theta}}(\mathcal{O}_{ia}^{ua}, \mathcal{Z}_{ia}^{ua})$, can be represented as:

$$
\begin{aligned}
p_{\boldsymbol{\theta}}(\mathcal{O}_{ia}^{ua}, \mathcal{Z}_{ia}^{ua}) &= p_{\boldsymbol{\theta}}(\mathbf{X}_{ia}, \mathbf{Z}_i^n, \mathbf{Z}_a^a, \mathbf{Y}_i^u) \\
&= p(\mathbf{Z}_i^n) p(\mathbf{Z}_a^a) p(\mathbf{Y}_i^u) p_{\boldsymbol{\theta}^a}(\mathbf{X}_{ia} \mid \mathbf{Z}_i^n, \mathbf{Z}_a^a, \mathbf{Y}_i^u),
\end{aligned} \tag{28}
$$

and the variational posterior $q_{\boldsymbol{\phi}}(\mathcal{Z}_{ia}^{ua} \mid \mathcal{F}_{ia}^{ua})$ can be factorized as:

$$
\begin{aligned}
q_{\boldsymbol{\phi}}(\mathcal{Z}_{ia}^{ua} \mid \mathcal{F}_{ia}^{ua}) &= q_{\boldsymbol{\phi}}(\mathbf{Z}_i^n, \mathbf{Z}_a^a, \mathbf{Y}_i^u \mid \mathbf{F}_i^n, \mathbf{F}_a^a) \\
&= q_{\boldsymbol{\phi}^n}(\mathbf{Z}_i^n \mid \mathbf{F}_i^n, \mathbf{Y}_i^u) q_{\boldsymbol{\phi}^a}(\mathbf{Z}_a^a \mid \mathbf{F}_a^a) q_{\boldsymbol{\phi}^c}(\mathbf{Y}_i^u \mid \mathbf{F}_i^n)
\end{aligned} \tag{29}
$$

Substituting Eq. 16 and Eq. 29 into Eq. 27, Eq. 27 can be written as:

$$
\begin{aligned}
\log p(\mathcal{O}_{ia}^{ua}) &\geq \mathbb{E}_{q_{\boldsymbol{\phi}}(\mathcal{Z}_{ia}^{ua} \mid \mathcal{F}_{ia}^{ua})}[\log p_{\boldsymbol{\theta}^a}(\mathbf{X}_{ia} \mid \mathbf{Z}_i^n, \mathbf{Z}_a^a, \mathbf{Y}_i^u)] - KL(q_{\boldsymbol{\phi}^n}(\mathbf{Z}_i^n \mid \mathbf{F}_i^n, \mathbf{Y}_i^u) \mid p(\mathbf{Z}_i^n)) \\
&\quad - KL(q_{\boldsymbol{\phi}^a}(\mathbf{Z}_a^a \mid \mathbf{F}_a^a) \mid p(\mathbf{Z}_a^a)) + \mathcal{H}(q_{\boldsymbol{\phi}^c}(\mathbf{Y}_i^u \mid \mathbf{F}_i^n)) \\
&= \mathbb{E}_{q_{\boldsymbol{\phi}^c}(\mathbf{Y}_i^u \mid \mathbf{F}_i^n)}[-\mathcal{B}(\mathbf{X}_{ia}, \mathbf{Y}_i^u)] + \mathcal{H}(q_{\boldsymbol{\phi}^c}(\mathbf{Y}_i^u \mid \mathbf{F}_i^n)) \\
&\triangleq -\mathcal{C}(\mathcal{O}_{ia}^{ua}),
\end{aligned} \tag{30}
$$

where $\mathcal{C}(\mathcal{O}_{ia}^{ua})$ is denoted as the ELBO of $\mathcal{O}_{ia}^{ua}$.

### A.3 The implementation details of SCAN

**The Inference Process**

As illustrated in Fig. 2, there are three types of latent random variables, i.e. $\mathbf{Z}^n$, $\mathbf{Z}^a$ and $\mathbf{Y}^u$, and our objective function of Eq. 9 involves the computations of expectation and KL-divergence over the three latent random variables. Similar to VAE [18], we assume all of the prior of latent variables to be multivariate normal distribution:

$$
p(\mathbf{Z}_i^n) = \mathcal{N}(\mathbf{0}, \mathbf{I}_D), p(\mathbf{Z}_a^a) = \mathcal{N}(\mathbf{0}, \mathbf{I}_D), \tag{31}
$$

where $\mathbf{I}_D$ is identity matrix of size $D$. We also use $\mathbf{Y}_i^u$ to represent the latent class variables of a unlabelled node $i$, within which the probability of $k$-th element being 1 obeys a categorical distribution defined as follows,

$$
p(\mathbf{Y}_{ik}^u = 1) = \text{Cat}(\mathbf{Y}_i^u \mid \boldsymbol{\pi}), \tag{32}
$$

where $\boldsymbol{\pi}$ is the parameter of the categorical distribution.

The standard VAE [18] replaces inference of variational latent variable parameters with inference neural networks. In our work, the variational distributions of nodes and attributes are chosen to be a Gaussian distribution $\mathcal{N}(\boldsymbol{\mu}, \boldsymbol{\sigma})$, whose mean $\boldsymbol{\mu}$ and covariance matrix $\boldsymbol{\sigma}$ are inferred by two different inference models parameterized by $\phi^n$ and $\phi^a$, respectively. Specifically, we set the variational distribution of nodes as:

$$
\mathbf{Z}_i^n \sim q_{\boldsymbol{\phi}^n}(\mathbf{Z}_i^n \mid \mathbf{F}_i^n, \mathbf{Y}_i) = \mathcal{N}(\boldsymbol{\mu}_{\boldsymbol{\phi}^n}, \boldsymbol{\sigma}_{\boldsymbol{\phi}^n}^2 \mathbf{I}), \tag{33}
$$

where $[\boldsymbol{\mu}_{\boldsymbol{\phi}^n}; \boldsymbol{\sigma}_{\boldsymbol{\phi}^n}^2] \in \mathbb{R}^{N \times 2D}$ is inferred through a two-layer Graph Convolutional Network (GCN) [37]. In Eq. 33, the label of node $i$, i.e. $\mathbf{Y}_i$, is set to be its real label $\mathbf{Y}_i^l$ if it is a labelled node, or its latent label $\mathbf{Y}_i^u$ inferred by label inference model $q_{\boldsymbol{\phi}^c}(\mathbf{Y}_i^u \mid \mathbf{F}_i^n)$ if it is an unlabelled node. In particular, the two-layer GCN is defined as:

$$
\begin{aligned}
\mathbf{H}_n^{(1)} &= \text{Tanh}(\tilde{\mathbf{A}} \mathbf{F}^n \mathbf{W}_n^{(0)}), \\
[\boldsymbol{\mu}_{\boldsymbol{\phi}^n}; \boldsymbol{\sigma}_{\boldsymbol{\phi}^n}^2] &= \tilde{\mathbf{A}} \mathbf{H}_n^{(0)} \mathbf{W}_n^{(1)},
\end{aligned} \tag{34}
$$

where $\mathbf{F}^n = [\mathbf{A}; \mathbf{X}]$ is the input feature of nodes, $\text{Tanh}(\cdot)$ is the hyperbolic tangent activation function, $\tilde{\mathbf{A}} = \mathbf{D}^{-\frac{1}{2}} \mathbf{A} \mathbf{D}^{-\frac{1}{2}}$ is the symmetrically normalized adjacency matrix with $\mathbf{D}_{ii} = \sum_j \mathbf{A}_{ij}$ being $\mathcal{G}$'s degree matrix, and $\phi^n = [\mathbf{W}_n^{(0)}; \mathbf{W}_n^{(1)}]$ are trainable parameters for the node inference layers, respectively.

For the variational distribution of attributes, we infer its parameters by a two-layer fully connected neural network, which is given as follows:

$$
\mathbf{Z}_a^a \sim q_{\boldsymbol{\phi}^a}(\mathbf{Z}_a^a \mid \mathbf{F}_a^a) = \mathcal{N}(\boldsymbol{\mu}_{\boldsymbol{\phi}^a}, \boldsymbol{\sigma}_{\boldsymbol{\phi}^a}^2 \mathbf{I}),
$$

$$\mathbf{H}_a^{(1)} = \mathrm{Tanh}(\mathbf{F}^a \mathbf{W}_a^{(0)} + \mathbf{b}_a^{(0)}),$$
$$[\boldsymbol{\mu}_{\boldsymbol{\phi}^a}; \boldsymbol{\sigma}_{\boldsymbol{\phi}^a}^2] = \mathbf{H}_a^{(1)} \mathbf{W}_a^{(1)} + \mathbf{b}_a^{(1)}, \tag{35}$$

where $\mathbf{F}^a = [\mathbf{X}^T]$ is the input feature of nodes and $\boldsymbol{\phi}^a = [\mathbf{W}_a^{(0)}; \mathbf{b}_a^{(0)}; \mathbf{W}_a^{(1)}; \mathbf{b}_a^{(1)}]$ are trainable parameters for the attribute inference layers, respectively.

The predictive label distribution $q_{\boldsymbol{\phi}^c}(\mathbf{Y}_i^u \mid \mathbf{F}_i^n)$ acts as the discriminative model trained for node classification according to the labelled nodes. The latent label distribution for unlabelled nodes is specified as categorical distribution over the $K$ classes:

$$\mathbf{Y}_i^u \sim q_{\boldsymbol{\phi}^c}(\mathbf{Y}_i^u \mid \mathbf{F}_i^n) = \mathrm{Cat}(\mathbf{Y}_i^u \mid \boldsymbol{\pi}_{\boldsymbol{\phi}^c}), \tag{36}$$

where $\boldsymbol{\pi}_{\boldsymbol{\phi}^c} = [\pi_1, \cdots, \pi_K]$ is inferred by the two-layer fully connected layer with a softmax output layer:

$$\mathbf{H}_c^{(1)} = \mathrm{Tanh}(\mathbf{F}^n \mathbf{W}_c^{(0)} + \mathbf{b}_c^{(0)}),$$
$$[\pi_1, \cdots, \pi_K] = \mathrm{softmax}(\mathbf{H}_c^{(1)} \mathbf{W}_c^{(1)} + \mathbf{b}_c^{(1)}), \tag{37}$$

where $\boldsymbol{\phi}^c = [\mathbf{W}_c^{(0)}; \mathbf{b}_c^{(0)}; \mathbf{W}_c^{(1)}; \mathbf{b}_c^{(1)}]$ are the parameters in the discriminative model for nodes' class labels.

**The Generative Process**

Suppose there are $K$ class labels, the observed data points, i.e. edges in adjacency matrix $\mathbf{A}$ and attributes in the node attribute matrix $\mathbf{X}$, are generated by the following process:

(1) For each unlabelled node $i$, draw label[1]:

$$\mathbf{Y}_i \sim \mathrm{Cat}(\mathbf{Y}_i^u \mid \boldsymbol{\pi}_{\boldsymbol{\phi}^c}), \tag{38}$$

where $\mathrm{Cat}(\cdot)$ is the Categorical distribution.

(2) For each node $i$ and each attribute $a$, draw latent vectors:

$$\mathbf{Z}_i^n \sim \mathcal{N}(\boldsymbol{\mu}_{\boldsymbol{\phi}^n}, \boldsymbol{\sigma}_{\boldsymbol{\phi}^n}^2 \mathbf{I}), \mathbf{Z}_a^a \sim \mathcal{N}(\boldsymbol{\mu}_{\boldsymbol{\phi}^a}, \boldsymbol{\sigma}_{\boldsymbol{\phi}^a}^2 \mathbf{I}). \tag{39}$$

(3) For each edge $\mathbf{A}_{ij}$ in adjacency matrix $\mathbf{A}$,

    (a) if $\mathbf{A}_{ij}$ is binary, draw:
$$\mathbf{A}_{ij} \sim \mathrm{Ber}(\boldsymbol{\mu}_{\boldsymbol{\theta}^e}), \tag{40}$$
    where $\mathrm{Ber}(\cdot)$ is Bernoulli distribution.

    (b) if $\mathbf{A}_{ij}$ is real, draw:
$$\mathbf{A}_{ij} \sim \mathcal{N}(\boldsymbol{\mu}_{\boldsymbol{\theta}^e}, \boldsymbol{\sigma}_{\boldsymbol{\theta}^e}^2 \mathbf{I}). \tag{41}$$

(4) For each attribute $\mathbf{X}_{ia}$ in adjacency matrix $\mathbf{X}$,

    (a) if $\mathbf{X}_{ia}$ is binary, draw:
$$\mathbf{X}_{ia} \sim Ber(\boldsymbol{\mu}_{\boldsymbol{\theta}^a}). \tag{42}$$

    (b) if $\mathbf{X}_{ia}$ is real, draw:
$$\mathbf{X}_{ia} \sim \mathcal{N}(\boldsymbol{\mu}_{\boldsymbol{\theta}^a}, \boldsymbol{\sigma}_{\boldsymbol{\theta}^a}^2 \mathbf{I}). \tag{43}$$

As all the edges and attributes in our experimental datasets are binary valued, we implement the generative model simply by inner product between latent variables, i.e.,

$$\boldsymbol{\mu}_{\boldsymbol{\theta}^e} = \mathrm{Sigmoid}(\langle [\mathbf{Z}_i^n; \mathbf{Y}_i], [\mathbf{Z}_j^n; \mathbf{Y}_j] \rangle), \tag{44}$$
$$\boldsymbol{\mu}_{\boldsymbol{\theta}^a} = \mathrm{Sigmoid}(\langle [\mathbf{Z}_i^n; \mathbf{Y}_i], \mathbf{Z}_a^a \rangle), \tag{45}$$

where $\langle \cdot, \cdot \rangle$ is the inner product operator, and $\mathrm{Sigmoid}(\cdot)$ is the sigmoid function. In the situation, to make the inner operation of Eq. 45 valid, we set the dimension of the latent attribute variable to be $K + D$.

## A.4 Details of the Datasets

We conduct experiments on four real-world attributed network datasets, statistics information of which is provided in Tab. 4:

- **Cora** & **Pubmed** [12]: These two datasets are citation networks, where nodes are publications and edges are citation links. Attributes of each node are bag-of-words representations of the corresponding publications.

- **BlogCatalog** [36]: This is a network dataset with social relationships of bloggers from the BlogCatalog website, where nodes' attributes are constructed by the keywords of user profiles. The labels represent the topic categories provided by the authors.

- **Flickr** [36]: It is a social network where nodes represent users and edges correspond to friendships among users. The labels represent the interest groups of the users.

Table 4: Statistics of the datasets in experiments.

| Datasets | #Nodes | #Edges | #Attributes | #Labels |
|---|---|---|---|---|
| **Pubmed** | 19,717 | 44,338 | 500 | 3 |
| **BlogCatalog** | 5,196 | 171,743 | 8,189 | 6 |
| **Flickr** | 7,575 | 239,738 | 12,047 | 9 |

## A.5 Complexity Analysis.

As shown in Appendix A.3, the layer-wise propagation rule for both the encoder and decoder networks is the main time cost of our algorithm, while the two-layer GCN network has the highest computational complexity in the computational propagation flow. The time complexity of the two-layer GCN network for one epoch boils down to 2 sparse-dense-matrix multiplications for a cost of $O(|\tilde{\mathbf{A}}^+|(H + D + M + N))$, where $|\tilde{\mathbf{A}}^+|$ denotes the number of nonzero entires in the Laplacian matrix, $H$ is the dimension of the first hidden layer, $D$ is the dimension of latent embeddings, $M + N$ is the dimension of node features. Empirically, our SCAN costs around 57s and 290s per 10 epochs on the Flickr and Pubmed datasets, respectively, for training on an Inter i7 3.60GHz CPU computer.

## A.6 Parameter Setting

We implement all the baselines with the codes released by the authors and tune the parameters of the baselines to be optimal. For all the comparison methods, the dimension of embeddings is set to be 64 unless specifically stated. We train our model by Adam optimizer with the learning rate being 0.01 in the iterations. The parameter $\beta$ of our SCAN model is set to be 0.5 in the node classification task, and is task-specific tuned to obtain the best performance for tasks of link prediction and attribute inference. The parameter $\alpha$ is also tuned to obtain the best performance for the node classification task.

## A.7 Parameter Sensitivity

In our SCAN, we have introduced a hyper-parameter $\beta$ to balance the reconstruction accuracy between edges and attributes to get a task-specific model. To verify the effect of parameter $\beta$, we conduct experiments on the BlogCatalog dataset: we evaluate RUC and AP scores of link prediction and attribute inference by different $\beta$, ranging from 0.1 to 0.9. The result is shown in Fig. 4(a). We can observe from Fig. 4(a) that, as $\beta$ increases, the performance of link prediction gets better and the performance of attribute inference gets worse. The result is quite intuitive as $\beta$ controls weight of reconstructing error between edges and attributes of the overall ELBO. This suggests the effectiveness of $\beta$ balancing the accuracy between link reconstruction and attribute inference and making our model to be task-specific.

To show the effectiveness of SCAN on semi-supervised node classification, we also evaluate the performance of our model on different ratio of labelled data. We conduct our experiment on BlogCatalog, where we set the other parameters to be optimal, vary the ratio of labelled data from

(a) Link predition and attribute inference performance by varying $\beta$.

(b) Node classification performance by varying label ratio.

Figure 4: Effect of parameters on BlogCatalog Dataset.

0.03 to 0.2 and evaluate the corresponding accuracy of SCAN_DIS, SCAN_SVM and Planetoid-T. We take Planetoid-T as our comparison model as it performs best in all the other baseline models on BlogCatalog dataset. The results are shown in Fig. 4(b). As shown in the figure, the classification accuracy of all the models increase as the number of labelled data increases. But we can notice that as the ratio gets higher, the discriminator of our model outperforms all the other models and achieves the best classification results. The SVM model trained using our learned representations consistently outperforms Planetoid-T and gets the second best classification accuracy. The result again demonstrates the effectiveness of the discriminator in our model on predicting the class of the nodes and capability of learning high-quality embeddings of nodes.