[Reviews · NeurIPS 2019]

Reviewer 1



Clarity I think that both the structure and readability of the paper are quite nice. The authors define very well their objective function based on the information provided for the Semi-Supervised Variational Auto-Encoders. They also support the presentation of the approach with figures illustrating the general outline of the methodology and the graphical model of the proposed method. Here, I would like to point out to some inconsistencies related to the notation used: - First of all, the paper does not contain line numbers since the authors did not omit the final and preprint options. - In the line just before Equation 1, it is stated that $N_1$ is the number of unlabeled data points, but in Equation 1, the upper bound of the summation containing the labeled data points is also $N_1$. I think the upper bounds of the summation terms should be replaced. - In Section 2, the word “generates” should be replaced with “generate” in the sentence starting with “More recently, a semi-supervised deep ...”. - In Subsection 4.1, the authors may replace the expression $\textbf{A} \in \mathbb{R}^{N\times N}$ with \textbf{A} \in \{0,1\}^{N\times N}$ to avoid the confusion since $\textbf{A}$ is the adjacency matrix. Originality The number of node representation learning approaches have been significantly increased in recent years but there are only a few semi-supervised methods targeting to attributed networks. As the authors in Subsection 3.1 claim, the paper might be the first approach to learn node representations along with the attribute embeddings in a semi-supervised manner. Quality From the technical point of view, the paper seems to be well-prepared. In particular, the authors present their approach with detailed mathematical derivations based on the Semi-Supervised Variational Auto-Encoders and they extensively evaluate the performance of the method in various downstream tasks. In the experiments, they report the results only for a chosen training size --- it could also be nice to examine the performance of the method for varying training set sizes. Significance As it was stated above, although there are many different types of approaches that learn node embeddings, this method seems to be the first one that learns representations of nodes and attributes in a semi-supervised manner.

Reviewer 2



- The authors present a semi-supervised graph embedding procedure that simultaneously embeds nodes and attributes into the same semantic space. The procedure leverages SVAE on heterogeneous data and several variational inference tricks for efficient implementation. Moreover, the node/attribute representations (being modeled via Gaussians in the SVAE) naturally have measures of uncertainty built their representations - Details in Section 3.2 are a bit confusing as presented. I believe you want $$\mathcal{O}=\mathcal{X}\times\mathcal{X}\times\mathcal{R}\times\mathcal{Y},$$ as you want observations to have two entities, a relation between the two entities, and possible class labels for the entities. As written, the definitions of $\mathcal{R}$ and $\mathcal{O}$ do not quite make sense. In addition, are the relations entity or class dependent; i.e., for $x_i\in X^g$ and $x_i\in X^h$ does $r_{ij}$ depend only on $g$ and $h$ or on the entities as well? Finally, I am unsure how you derived (4)? - How scalable is SCAN? It would be helpful to know the orders of the 3 data sets in the experiments and what the runtime of SCAN was in each setting. - How are you choosing D? - In the Attribute Inference experiment section, is SCAN making use of label semi-supervision? Are the algorithms being compared to unsupervised? The experiments would benefit from more experiments on topologically diverse networks to show the effectiveness of your approach in a variety of real data settings. - The ellipsoids in Figure 4 are too small to be easily observed. - There are a number of grammatical errors/quirks in the manuscript which impact overall readability. For example: Abstract, line 2: offers -> offer Abstract, line 7: remove "the" Page 1, line -3: A number of work has -> A number of works have Page 2, line 6: has -> have Page 2, line 6: profound probability theoretical basis -> profound basis in theoretic probability Page 2, line 8: offers -> offer Page 2, line 9: has -> have Page 2, 2nd full paragraph, line 5: consisted -> consisting etc...

Reviewer 3



This paper proposes a novel semi-supervised co-embedding model for attributed networks, based on Semi-supervised VAE. The model design is reasonable, by considering the dependency of node-node and node-attribute in five different cases. The inference process also makes sense. It improves the existing unsupervised co-embedding model by learning also with partially labeled nodes in a semi-supervised way. The introduced model outperforms the state-of-the-art attributed graph embedding models. The paper is also well written and easy to follow. There are some writing errors to correct and some unclear notations to clarify. Please see the “improvements” part. Another suggestion is about the evaluation of the performance on different ratio of labelled data. It will be better to include it in the main text, than leaving it in the supplementary document, because the semi-supervised learning models should be evaluated on the capability of learning from different small portions of labeled data.

[Author Response · NeurIPS 2019]

We would like to thank all the reviewers for your helpful comments and suggestions. We appreciate your positive
comments on our work: "nice structure and readability", "well-written" and "reasonable design". Hereafter, we first
provide two responses to the common concerns raised by the reviewers, and then reply each reviewer, respectively.

**Common Response 1: Complexity analysis of our SCAN model.** As shown in Appendix A.3, the layer-wise
propagation rule for both the encoder and decoder networks is the main time cost of our algorithm, while the two-layer
GCN network has the highest computational complexity in the computational propagation flow. The time complexity
of the two-layer GCN network for one epoch boils down to 2 sparse-dense-matrix multiplications for a cost of
$O(|\tilde{\mathbf{A}}^+|(H + D + M + N))$, where $|\tilde{\mathbf{A}}^+|$ denotes the number of nonzero entires in the Laplacian matrix, $H$ is the
dimension of the first hidden layer, $D$ is the dimension of latent embeddings, $M + N$ is the dimension of node features.
Empirically, our SCAN costs around 57s and 290s per 10 epochs on the Flickr and Pubmed datasets, respectively, for
training on an Inter i7 3.60GHz CPU computer.

**Common Response 2: Representation updates on Section 3.2.** Considering the comments of Reviewer #2 and
Reviewer #3, we now revise $\mathcal{O} = (\mathcal{X}, \mathcal{Y}, \mathcal{R})$ to be $\mathcal{O} = ((\mathcal{X}^g, \mathcal{Y}^g) \times (\mathcal{X}^h, \mathcal{Y}^h) \times \mathcal{R}^{gh}), g, h \in \{1, \cdots, T\}$, which
is the generalized heterogeneous data in our paper, where the relationship type $\mathcal{R}^{gh}$ is depend on the entity types
$g$ and $h$. In our task, we have two entity types and two relationship types, which can be represented as $\mathcal{O}_{AN} =$
$((\mathcal{X}^1, \mathcal{Y}^1) \times (\mathcal{X}^1, \mathcal{Y}^1) \times \mathcal{R}^{11}, (\mathcal{X}^1, \mathcal{Y}^1) \times (\mathcal{X}^2) \times \mathcal{R}^{12})$. We will revise the corresponding representations in Section 3.2
based on this formulation and focus more on the specific heterogeneous data type in our task (i.e. $\mathcal{O}_{AN}$). For instance,
Eq (4) will be revised to $p_\theta(\mathcal{O}_{ij}, \mathcal{Z}_{ij}) = p_\theta(r_{ij}^{gh} \mid \mathcal{Z}_{ij}, \mathbf{Y}^l)p(\mathbf{Y}^l) \prod_{\mathbf{z} \in \mathcal{Z}_{ij}} p(\mathbf{z})$. We will further provide a Section to
discuss the potential extension of our model to handle other types of heterogeneous data, such as bipartite graph in
recommender system, multi-relationship in knowledge graph and multi-type of entities; and clarify the difference on the
variation and the capability of our model when handling these type of heterogeneous data.

**Response to Reviewer #1**

- Thank you for reviewing our paper and noticing the typos and improper notations in our paper. We have double
checked the whole paper and corrected all the typos we can find.
- Regarding the performance on varying training set sizes, we have conducted experiments. Here we only show the
accuracy result on the BlogCatalog dataset, and detailed analysis will be provided in the Appendix of our final version.

27
| % training | 85% | 80% | 75% | 70% | 65% | 60% | 55% | 50% |
|---|---|---|---|---|---|---|---|---|
| SCVA_DIS | .845 | .838 | .833 | .828 | .810 | .792 | .780 | .778 |

- Regarding the complexity analysis of our model, please see the response in **Common Response 1**.

**Response to Reviewer #2**

- Thank you for reviewing our paper and noticing the typos and confusing definition in our paper. We have double
checked the whole paper, corrected all the typos we can find and enlarged Figure 4 to make it more legible.
- Thank you for the suggestions on Section 3.2. Please see the response in **Common Response 2**.
- Eq (4) is a generalized form of the factorization of the joint probability with two entities, their labels and their relations,
where we assume that the labels of two entities are not explicitly given in current form. We now modified Eq (4) to make
it more clear (See **Common Responose 2**), where the full derivation will be in final version. The specific factorization
forms on different cases can be found in the Appendix (Please see Eq (12), Eq (16), Eq (20), Eq (24) and Eq (28)).
- We now provide the complexity analysis of our model; please see the response in **Common Response 1**. We will
also provide the statistics information of the 3 datasets and report the the runtime of SCAN with each setting in the
Appendix of our final version.
- We tuned the latent dimension $D$ in $\{16, 32, 64, 128\}$, and chosen $D$ with a best performance on the validate sets in
these tasks. In our paper, $D = 64$ was a default setting, with which the best performance is achieved in all the datasets.
- In the attribute inference task, our SCAN did make use of the label information to obtain their embeddings. We indeed
have made comparison with state-of-the-art unsupervised methods (such as CAN) in terms of node classification task,
and our model shows better performance. For fair comparison we only report the result on semi-supervised task.
- Thanks for the good suggestions. We will move the performance result over different label ratio to the main text, and
seek to evaluate our proposed model on more topologically diverse real datasets and report result in the final version.

**Response to Reviewer #3**

- Thank for the good comment on "little over-claimed". Please see the response in **Common Response 2**.
- The title of Section 3.2 will be revised to "Semi-supervised Learning for Heterogeneous Data".
- Thank you for noticing the typos. In Eq (5), $\mathbf{Y}^l$ in $q_\phi$ should be omitted, and log should be added to $p(z)p(y)$.
- There is no sum on $p_\theta$, since $\mathcal{O}_{ij}$ is an atomic data point. Please see the revised form of Eq (4) in **Common Response**
**2**, where the full factorization will given in the appendix of the final version. $q_\phi$ can be factorized in different forms
according to different cases (Please see Eq (13), Eq (17), Eq (21), Eq (25) and Eq (29) in the appendix of the paper).

[Meta-Review · NeurIPS 2019]

The paper tackles the problem of embedding partially labelled attributed networks. It proposes a semi-supervised co-embedding model for attributed networks by generalising the SVAE framework to deal with heterogeneous data. All three reviewers agree that the paper is well written, the methodology is novel and would make a good NeurIPS paper. The empirical evaluation was considered sufficient. The author's rebuttal was very clear and addressed most of the reviewers' concerns.